# Identification of Aging and Young Subtypes for Predicting Colorectal Cancer Prognosis and Immunotherapy Responses

**DOI:** 10.3390/ijms24021516

**Published:** 2023-01-12

**Authors:** Lulu Tan, Xiakeerzhati Xiaohalati, Feng Liu, Jia Liu, Haoyu Fu, Yang Zhang, Jinbo Gao, Kaixiong Tao, Guobin Wang, Lin Wang, Zheng Wang

**Affiliations:** 1Department of Gastrointestinal Surgery, Union Hospital, Tongji Medical College, Huazhong University of Science and Technology, Wuhan 430022, China; 2Department of Clinical Laboratory, Union Hospital, Tongji Medical College, Huazhong University of Science and Technology, Wuhan 430022, China; 3Research Center for Tissue Engineering and Regenerative Medicine, Union Hospital, Tongji Medical College, Huazhong University of Science and Technology, Wuhan 430022, China

**Keywords:** colorectal cancer, aging, unsupervised clustering, tumor microenvironment, prognosis

## Abstract

Colorectal cancer (CRC) is critically related to aging and severely threatens human lives. To better explore the effects of aging on CRC progression and therapy outcome, a reliable aging subtypes identification of CRC is urgently desired. Here, 28 aging-related genes associated with the CRC prognosis were selected by univariate Cox analyses. Based on these 28 genes, CRC patients were divided into the aging subtype and young subtype by non-negative matrix factorization clustering. Aging subtype and young subtype of CRC were identified with distinct molecular features and clinical prognosis. The aging subtype was characterized by upregulation of senescence-associated secretory phenotype, higher frequencies of TP53 and immune checkpoint molecules, and high sensitivity to protein kinase and angiogenesis inhibitors. Furthermore, 14 genes were selected by LASSO penalized Cox regression analyses for aging-related risk signature construction. The constructed aging risk signature exhibited good prediction and the nomogram showed robust discrimination power over the traditional CRC staging system. In conclusion, this study successfully established aging subtype and young subtype of CRC, which is helpful to identify patients with aging characteristics to evaluate prognosis and treatment outcomes. Introducing aging-based subtypes into clinical concern and patient prognostication provides new opportunities for personalized CRC treatment.

## 1. Introduction

Colorectal cancer (CRC) is the second most deadly cancer worldwide [1]. CRC treatments include surgical resection combined with chemotherapy, radiotherapy, and immunotherapy. Although this combination therapy achieved relatively long survival rates, the benefit was limited to a small portion of patients. Many patients still suffer recurrence and metastasis due to the high molecular heterogeneity of CRC [2]. Despite the traditional tumor–node–metastasis (TNM) staging system and conventional biomarkers, cancer sub-classification based on pathological characteristics and gene expression features has been increasingly developed for accurate prognosis assessment and personalized clinical management [3,4]. The consensus molecular subtype (CMS) is the most reliable classification system currently available for CRC subtyping [5]. In addition, several CRC subtypes have been identified to predict prognosis and therapeutic responses, including hypoxia-related subtypes [6], metabolism-related subtypes [7], pyroptosis-related subtypes [8], and immune-related subtypes [9]. These tumor subtype classification strategies provide accurate prediction tools for cancer prognosis, and broaden the targeted therapy options. 

Aging is an important physiological process of living organisms, and it has been reported to have pleiotropic effects on cancer promotion [10]. Studies have shown that senescent cells have highly enhanced pro-survival and anti-apoptotic abilities to facilitate cancer development [11]. Moreover, senescent cells are thought to mediate cancer invasion and metastasis by secreting the senescence-associated secretory phenotype (SASP) [12]. Senescent cells remodel the tumor microenvironment (TME) of multi-types of tumors, including lung adenocarcinoma and CRC, resulting in poor prognosis [13,14]. In addition, senescent tumor cells are involved in immunotherapy resistance and become therapeutic targets in CRC therapies [15]. Recently, Yue et al. demonstrated that aging is an independent prognostic factor of CRC and an aging-based risk signature can predict the survival of CRC patients [16]. However, there is no reliable aging subtype classification of CRC to facilitate personalized treatment.

In this study, two aging-related CRC subtypes (aging subtype and young subtype) were distinguished. Their subtype-specific molecular changes, immune characteristics, chemotherapy response, and survival stratification were comprehensively analyzed. Furthermore, gene signature and clinicopathological features were integrated to develop a risk model with high prognostic accuracy and predictive power, offering an alternative approach for CRC prediction. Our findings throw light on the prognostic significance of aging in CRC and contribute to the emerging field of aging-related CRC classification and targeted therapy.

## 2. Results

### 2.1. Aging-Related Genes-Based Clustering and Subtypes Identification

A total of 279 aging-related genes were acquired from the CellAge database. By univariate Cox regression analysis, we obtained 28 genes associated with CRC prognosis, of which 11 genes were protective factors, whereas the rest were risk factors (Appendix A). The associations of these genes were identified based on the PPI network that was explored using the STRING database and visualized by Cytoscape (Appendix A). 

The non-negative matrix factorization (NMF) was applied to identify the CRC subtypes based on aging-related genes (Figure 1a,b). The results showed that the clustering effect was optimal when CRC patients were divided into two subgroups with good internal consistency and stability, which was further confirmed by principal component analysis (PCA) (Figure 1c). Similarly, CRC patients can be classified into two subgroups by consensus clustering (Figure 1d–f). In addition, there was a statistically significant difference between CRC patients in the two cluster groups in terms of T stage, N stage, M stage, clinical stage, lymphatic invasion, venous invasion, and alive outcome (Figure 1g and Table 1). Cluster 2 shows better prognosis than Cluster 1 according to survival analysis. Similar results were revealed in an independent dataset (Appendix A).

We also found that Cluster 1 was related to higher expression of SASP (Figure 1i and Appendix A). In addition, gene set enrichment analysis (GSEA) showed that the aging-related gene sets were significantly activated in Cluster 1 (Appendix A). Considering the above results, two CRC subgroups were identified. Cluster 1 was designated as the aging subtype, and Cluster 2 was thereafter designated as the young subtype based on the relative downregulation of aging-related genes.

### 2.2. Clinical Characteristics of Aging-Related Genes-Based Subtypes

We next compared the clinical outcomes between the developed two subtypes using stratified survival analysis. We found that two subtypes showed significant differences in the OS of CRC patients subgrouped by age, stage, T, N, and M (Figure 2a). Moreover, univariable and multivariable Cox regression analysis showed that aging was an independent and significant risk prognostic factor for survival in patients with CRC (Figure 2b,c). Together, our evidence suggests that aging-based clustering has excellent predictive power for the prognosis of CRC patients, while aging subtypes of CRC patients have an inferior prognosis.

### 2.3. The Difference in Molecular Characteristics between Aging-Related Genes-Based Subtypes

Considering the crucial role of genomic alterations in the progression of CRC, we also investigated genetic mutation differences between two subtypes. As shown in the mutation map, the top three mutational genes in both subtypes were TP53, APC, and TTN (Figure 3a,b). TP53 was frequently mutated in the aging subtype, whereas APC was frequently mutated in the young subtype. Furthermore, gene set variation analysis (GSVA) was introduced to uncover the potential pathways involved in the two subtypes (Figure 3c). The EMT, ANGIOGENESIS, and HEDGEHOG_SIGNALING pathways were mainly enriched in the aging subgroup. Meanwhile, the “cell cycle”-associated pathways (e.g., E2F_TARGETS and G2M_CHECPOINT) were enriched in the young subgroup (Figure 3d). Similar results were observed in the GSEA analysis in the TCGA database subsequently. Moreover, the aging subgroup was resistant to typical cell cycle inhibitors and sensitive to most protein kinase inhibitors and angiogenesis inhibitors (Figure 3e). These findings suggest that care should be taken when using chemotherapeutic agents in patients with different subtypes of CRC.

### 2.4. Aging-Related Genes-Based Subtypes Show Different Immune Features

Given that the infiltration of immune cells is the crucial factor affecting tumor progression and immunotherapy response, we estimated the differences in immune cell infiltration of TME between the two subtypes. The stromal, immune, and estimate score of aging subtypes was higher than that of the young subtype (Figure 4a). Plasma cells, activated memory CD4^+^ T cells, and activated DC cells were significantly upregulated in the young subtype. Regulatory T cells (Tregs), M0 macrophages, endothelial cells, and fibroblasts were markedly upregulated in the aging subtype (Figure 4b and Appendix A). Notably, as shown in Figure 4c, a series of immune checkpoints were widely upregulated in the aging subtype. Furthermore, compared with the young subtype, the TIDE score for immunotherapy and microsatellite instability value were elevated in the aging subtype (Figure 4d). Conversely, the IPS score of the aging subtype was lower than that of the young subtype (Figure 4e). In conclusion, the estimate score, TIDE score, and IPS score together demonstrated the insensitivity of the aging subtype to immunotherapy.

### 2.5. Aging-Related Gene Signature Is a Prediction Tool for CRC Prognosis

Differential expression genes (DEGs) were profiled by volcano plot analysis (Figure 5a). The GO enrichment analysis showed that the DEGs were enriched in ECM–receptor interaction, Malaria, and PI3K-Akt signaling pathway (Figure 5b). The LASSO penalized Cox regression analysis was utilized to construct the aging risk signature of CRC patients from DEGs (Figure 5c,d). A total of 14 aging-related genes (CLCA1, FBXO16, HOXC6, HOTAIR, HOXC8, HOXC11, KCNQ2, MUC16, NOG, NKAIN4, PCOLCE2, PANX2, SULT1B, and TNNT1) were finally selected, and were introduced to establish a novel risk score (Figure 5e). ROC analysis showed that the area under the curve was 0.851 (Figure 5f), indicating that our risk signature can accurately distinguish the aging and young subtype. GSEA found that the expression of the aging group genes was positively enriched in GOBP_AGING, GOBP_CELL_AGING, and CELLULAR_SENESCENCE (Figure 5g).

Then, we calculated the aging risk signature score of CRC patients based on the expression levels of 14 genes and patients were divided into a high risk group and a low risk group (Figure 5h). Kaplan–Meier analysis revealed that CRC patients with higher risk scores had a poor prognosis (Figure 5i). More importantly, our aging-related gene signature had good prognostic accuracy indicated by the time-dependent ROC analysis (Figure 5j). The same results were observed in two independent data sets (Appendix A).

### 2.6. Aging-Based Nomogram Improves Survival Prediction for CRC

To validate the prognostic value of the aging signature in five different independent cohorts, we integrated the survival outcomes from GSE39852, GSE16158, GSE17536, GSE38832, and GSE17537 cohorts. As shown in Figure 6a, a positive correlation between aging and overall mortality was observed in CRC patients, suggesting that the aging signature is a prognostic factor in different cohorts. At last, we constructed a nomogram with five independent prognostic factors to provide a quantitative tool for OS prediction of CRC patients, and used the nomogram to calculate the total scores of individuals (Figure 6b). The nomogram exhibited high consistencies between the observed and predicted survival probability (Figure 6c).

In addition, the time-dependent AUC values of the nomogram for predicting 1-, 3-, and 5-year OS were 0.838, 0.836, and 0.804 (Figure 6d), respectively, indicating the favorable discrimination ability of this nomogram. Encouragingly, the prediction ability of this nomogram was better than that of the stage and risk score (Figure 6e).

### 2.7. Validation of Aging-Related Genes

Studies have demonstrated that upregulation of P21, CCL2, MMP-1, and downregulation of LMNB1 are common markers of cellular senescence [17]. To validate the expression patterns of aging-related genes in colon cancer cell lines, we compared the mRNA expression levels of these senescence markers between the LoVo, DLD1, SW48, and SW620. We found that the mRNA expression levels of P21, CCL2, and MMP-1 in LoVo were higher than those of DLD1, SW48, and SW620. The opposite results were observed in LoVo for LMNB1 expression (Figure 7a). We hypothesized that LoVo cells were relatively senescent compared with other cell lines. As shown in Figure 7b, the mRNA expression levels of KCNQ2, PCOLCE2, and TNNT1 in LoVo cells were significantly increased, while the expression levels of protective factors SULT1B1 and FBXO16 were decreased (Figure 7b), which is in line with the above results in Figure 5e. At the single-cell level, FBXO16 and SULT1B1 were mainly expressed in epithelial cells. TNNT1 was mainly expressed in endothelial cells, epithelial cells, macrophages, and monocytes. PCOLCE2 was mainly expressed in smooth muscle cells, and KCNQ2 was primarily expressed in endothelial cells, suggesting that these cell types may mainly contribute to the senescence in CRC (Figure 7c and Appendix A).

## 3. Discussion

In vertebrates, aging is a universal physiological feature of organisms that is characterized by the degeneration and functional decline of tissues and cells. Cellular senescence is historically considered an anti-tumor mechanism in response to oncogenic stress. Recently, studies suggest that senescent cells also create an immunosuppressive, vascularized, and pro-inflammatory tumor microenvironment through SASP, thereby promoting tumor growth and progression [18]. An increasing number of studies have found that the expression features of aging-related genes could predict the survival and treatment response of cancer patients, including CRC. However, whether CRC patients can be classified into different subtypes based on aging-related genes and understand their distinct characteristics for better clinical management has not yet been proposed. Hence, the further identification of aging-related subtypes may help to explain the aging heterogeneity among CRC patients and pre-recognize patients with aging features, thus providing precise treatment options.

Herein, we first recognized two distinct aging-related subtypes, which are named the aging subtype (Cluster 1) and the young subtype (Cluster 2) according to the expression status of aging-related genes. The two subtypes showed different clinical outcomes, molecular characteristics, tumor microenvironment, and drug sensitivity. We found that patients in the aging subtype have an inferior prognosis, immune suppression features, and low immunotherapy responsiveness. Subsequently, an aging risk signature was established for prognostic prediction for CRC patients and its reliable predictive ability was validated in five independent cohorts. The aging-based nomogram showed outstanding advantages in predicting the prognosis of CRC patients compared with the traditional prediction tools. 

Studies suggest that senescent cells secrete a spectrum of SASP factors, including cytokines, matrix proteases, chemokines, growth factors, and receptors, which reshape the tumor microenvironment and disrupt tissue homeostasis by affecting neighboring cells, inducing immune cell recruitment and pro-inflammation. Consistent with this, we noticed that patients in the aging subtype had a higher expression of SASP, including a series of interleukins (IL-6, IL-13, and IL-15), chemokines (CCL3, CCL8, CCL13, and CCL26), matrix proteases (MMP-1, MMP-9, MMP-13, and MMP-14), receptors or ligands (TNFRSF1A, TNFRSF1B, and PLAUR), etc. The pleiotropic effect of SASP can facilitate tumor invasion and metastasis. For example, inflammatory SASP such as CCL3 and IL-6 remodels the TME by recruiting immunosuppressive cells, thus protecting tumor cells from immunosurveillance [19,20]. MMPs can degrade the extracellular matrix to mediate CRC metastasis [21]. Thus, we hypothesized that the high levels of SASP may be one of the reasons for the poor prognosis of patients in aging subtypes.

Consensus molecular subtypes (CMS), the most robust classification system, divide CRC into four subtypes. CMS1 has a high BRAF mutation rate; CMS2 has a high APC mutation rate, and the WNT/MYC signaling pathway is significantly activated. CMS3 was characterized by a high frequency of KRAS mutations. CMS4 is characterized by high TP53 mutations. We found distinct genomic alterations in two subtypes, and TP53 mutation was more frequent in the aging subtype, which was similar to the characteristics of CMS4. As a key transcription factor, TP53 participates in cellular senescence through various signaling pathways. For example, the p53/p21 pathway is involved in cell cycle arrest and tumor suppression [22]. Hence, TP53 mutation in the aging subtype may lead to loss of the tumor-suppressive function and promote CRC progression. In the young subtype, APC mutation frequency was higher and the MYC signaling pathway was enriched and had a favorable prognosis, which was similar to the characteristics of CMS2. However, no significant differences were found in BRAF mutations and KRAS mutations between the aging and the young subtype.

The aging subtype was enriched in pathways related to the tumor development and metastasis, such as EMT, angiogenesis, and Notch signaling pathways. Studies have shown that these signaling pathways are involved in aging. For instance, EMT-inducing transcription factors are involved in the regulation of p53 and Bcl-2 expression, thus affecting cellular senescence [23]. In addition, the activated angiogenesis pathway has been observed in aging patterns of various cancer types, such as CRC and LUAD [14,15]. Moreover, the Hedgehog signaling pathway is dysregulated in senescent cells and often reactivated in cancer [24]. This may explain the inferior prognosis of patients in the aging subtype. EMT plays a pivotal role in tumor invasion and metastasis [25]. Senescent cells may influence EMT through secretion of SASP. For example, IL-6 has been reported to facilitate tumor invasion by disrupting cell adhesion [26,27]. In return, epithelial transformed mesenchymal cells degrade the extracellular matrix by highly expressed MMPs, a type of SASP, thereby enhancing tumor invasion [28]. Moreover, it is known that pathological angiogenesis provides nutritional support for tumor growth and is related to the distant metastasis of tumor cells [29]. Consistent with the enrichment of the angiogenesis signaling pathway, we found high expression of pro-angiogenic SASP factors such as IL-6 and VEGF in the aging subtype. These changes may promote CRC metastasis. Additionally, the aging subtype was also positively correlated with a Hedgehog signaling pathway whose non-canonical activation has been shown to be associated with cancer progression [30]. As expected, cell-cycle-related signaling pathways such as E2F_TARGETS and G2M_CHECKPOINT were negatively correlated with the aging subtype. More importantly, the aging subtype was insensitive to treatment with cell-cycle-specific anti-tumor drugs such as gemcitabine and etoposide, which was attributed to the cell-cycle arrest features of senescent cells. In addition, the aging subtype was more sensitive to angiogenesis and protein kinase inhibitors including pazopanib and imatinib. We hypothesized that this might be correlated with the enrichment of the above signaling pathways in the aging subtype. Our findings reveal that the patients of the aging subtype of CRC may not benefit from the cell cycle inhibitors but from antiangiogenic agents. Overall, this aging-based subtype classification can provide useful guidance for precise treatment.

Senescent cells establish the immunosuppressive microenvironment through a variety of mechanisms, including inducing the accumulation of immunosuppressive cells, and the secretion of immunosuppressive SASP factors, so as to evade the immune surveillance and anti-tumor immune response of the host. The two aging-related subtypes have different immune cell infiltration profiles. Activated memory CD4^+^ T cells and activated dendritic cells were significantly upregulated in the young subtype. Activated memory CD4^+^ T cells induce a robust anti-tumor response in the early stage of tumor progression [31]. Dendritic cells, as effective antigen-presenting cells, are generally associated with better tumor prognosis [32,33]. M0 macrophages and regulatory T cells (Tregs) were enriched in the aging subtype. In this study, we did not observe significant differences in M1 and M2 macrophage infiltration between the aging and young subtypes. However, M0 macrophages, the precursors of M1 and M2 macrophages, were highly infiltrated in the aging subtype. Evidence suggests that the accumulation of M0 macrophages in tumors is associated with a worse prognosis [34,35]. However, the clinical significance of M0 macrophage in the aging subtype of CRC still needs further study [36]. Tregs contribute to tumor development by hindering effective anti-tumor-specific immune responses in CRC patients [37]. Excessive accumulation of Tregs reduces the efficacy of immunotherapy and is often associated with poor prognosis [38]. Patients in the aging subtype showed higher infiltration of Tregs and overexpression of immune checkpoints, such as PD-L1, CTLA-4, and TIM3, indicating that TME was strongly inhibited. Furthermore, patients in the aging subtype had higher TIDE scores, suggesting the high immune escape potential of the aging subtype. The lower IPS scores confirmed the low immunogenicity of the aging subtype, which is consistent with the TIDE score prediction. Taken together, the above results suggest that the aging subtype with immunosuppression characteristics is mainly manifested by an increase in Tregs and immune checkpoints, which may be the reason for the low response to immune checkpoint inhibitor (ICI) therapy. On the contrary, patients in the young subtype are more likely to benefit from ICI therapy. These findings highlight the promoting role of aging-based subtype classification in personalized ICI therapy.

Some limitations should be acknowledged in the present study. First of all, the predictive reliability and clinical application of the aging risk signature need further validation in prospective clinical trials. Second, the predictive roles of aging-related genes determined in our study need to be verified in larger clinical samples. Last but certainly not least, fundamental experiments are urgently needed to elucidate the biological function of aging in tumorigenesis and development of CRC.

## 4. Materials and Methods

### 4.1. Data Preprocessing

A total of five available CRC gene expression datasets (GSE39582, GSE16158, GSE17536, GSE38832, and GSE17537) and relevant clinical information were downloaded from the Gene Expression Omnibus (GEO) (https://www.ncbi.nlm.nih.gov/geo/, accessed on 20 April 2022). The RNA-sequencing and somatic mutation data of TCGA-COAD were obtained from the UCSC public (https://xenabrowser.net/, accessed on 20 April 2022). Patients without specific survival data were excluded. The FPKM values of TCGA-COAD were transformed into transcripts per kilobase million (TPM) for subsequent analyses. The somatic mutation data of TCGA-COAD patients were analyzed by R package “maftools”. All expression profiles were normalized and log2 transformed as previously reported [39].

### 4.2. Unsupervised Clustering Analysis of Aging-Related Genes

The latest summary of 279 aging-related genes was obtained from the CellAge database (https://genomic.senescence.info/cells/, accessed on 20 April 2022). Then, 28 aging-related genes were selected via univariate Cox analysis. On the basis of the expression level of these 28 genes, 430 CRC from TCGA were classified using the Nonnegative Matrix Factorization (NMF) clustering analysis. To verify the accuracy of the clustering, the consensus clustering algorithm was also used for unsupervised clustering analysis. Ultimately, unsupervised clustering identified two subtypes. The dataset GSE39582 with larger sample size and more complete clinical information was used to validate the clustering results.

### 4.3. Clinicopathological Features between the Aging and Young Subtypes

To examine the clinical value of the two subtypes identified by unsupervised clustering, a chi-square test was used to demonstrate the relationship between molecular subtypes and clinicopathological features. Clinicopathological features included age, gender, survival status, lymphatic invasion, venous invasion, and tumor stage.

### 4.4. Immune Infiltration Analysis

To reveal the distinct tumor microenvironment between the aging and young subtypes, two algorithms named CIBERSORT [40] and MCP-counter [41] were used to quantify the relative or absolute abundance of immune cell populations in CRC. The estimate package predicts the amount of mesenchymal cells and immune cells in malignant tumor tissues. This scoring system is based on a single sample gene set enrichment analysis and generates three scores: stromal score, immune score, and estimate score [42]. The “estimate” R package was used to evaluate each patient′s immune score and stromal score.

### 4.5. Differentially Expressed Genes (DEGs) Identification

The DEGs between the aging and young subtypes were identified using the “limma” R package. Genes with an adjusted *p* value < 0.05 and a fold change of 1 were considered significantly different. Functional enrichment analysis of DEGs was performed using the David database (https://david.ncifcrf.gov/, accessed on 5 May 2022).

### 4.6. Construction of Aging-Related Gene Signature

To quantify the aging modification patterns of each patient, the aging score was calculated to assess all individuals with CRC. Univariate Cox regression analysis was performed on DEG to identify genes associated with CRC overall survival (OS). Based on aging-related prognostic genes, risk scores were calculated using 10-fold cross-validated least absolute shrinkage and selection operator (LASSO) regression. The 14 genes and their correlation coefficients were used to construct an aging gene signature, defined as an aging score. CRC patients were divided into high-risk and low-risk groups based on the median aging score. Survival analysis of high- and low-risk groups was performed using the Kaplan–Meier method. The receiver operating characteristic (ROC) curve was used to evaluate the ability of survival prediction.

### 4.7. Quantitative Real-Time Polymerase Chain Reaction PCR (RT-qPCR)

Total RNA was extracted from four colon cancer cells (DLD-1, LoVo, SW48, and SW620) using TRIzol reagent (Invitrogen, Thermo Fisher Scientific, Waltham, MA, USA). Total RNA was reverse transcribed to cDNA using a RT reagent kit (Vazyme Biotech, Nanjing, China). The RT-qPCR was performed using a SYBR-Green assays (Vazyme Biotech, Nanjing, China) on a StepOnePlusTM real-time PCR instrument (Thermo Fisher Scientific, Inc., USA). The mRNA expression level of P21, CCL2, MMP-1, LMNB1, KCNQ2, NOG, PCOLCE2, NKAIN4, HOXC6, HOTAIR, HOXC8, PANX2, HOXC11, TNNT1, MUC16, CLCA1, SULT1B1, and FBX016 was normalized with GAPDH and the data were calculated through the 2−ΔΔCt method. The primer sequences are listed in Appendix A.

### 4.8. Single-Cell RNA-Seq Data Analysis

The single-cell data of CRC were obtained from GSE132465. The raw data were converted into Seurat objects using the Seurat R package and filtered according to the following criteria: cells with >1000 unique molecular identifier (UMI) counts; >200 genes and <6000 genes; and <20% of mitochondrial gene expression. After batch effect correction based on the CCA, the cells were clustered into different subgroups by t-SNE projection, and cell types were annotated by SingleR. Subsequently, the expression of aging-related genes in different cell types was analyzed.

### 4.9. Additional Bioinformatic and Statistical Analysis

Principal component analysis (PCA) was used to visualize the differences between different groups. The “GSVA” and “GSEA” packages were used to analyze pathway enrichment between the two subtypes. The “pRophetic” package was used to predict half-inhibitory concentration (IC50) values of CRC therapeutics. Tumor immune dysfunction and exclusion (TIDE) [43] and immunophenoscore (IPS) [44] algorithms were used to predict potential immune checkpoint blockade treatment response. All statistical analyses were performed using R version 4.0.3. The Wilcoxon test was used for special variables (risk score, aging-related genes, and aging-related gene clusters) between the two groups. The chi-square test was used for categorical variables between the two groups. *p* < 0.05 was considered to be statistically significant. 

## 5. Conclusions

In conclusion, this study first identified two aging-related subtypes of CRC with different clinical outcomes, transcriptome characteristics, immune status, and treatment response. A novel aging risk model contributes to the accurate assessment of the prognosis of individual CRC patients. More importantly, the subtypes classification of CRC based on aging may provide a promising direction for therapeutic decision making in a clinical setting.

## Figures and Tables

**Figure 1 ijms-24-01516-f001:**
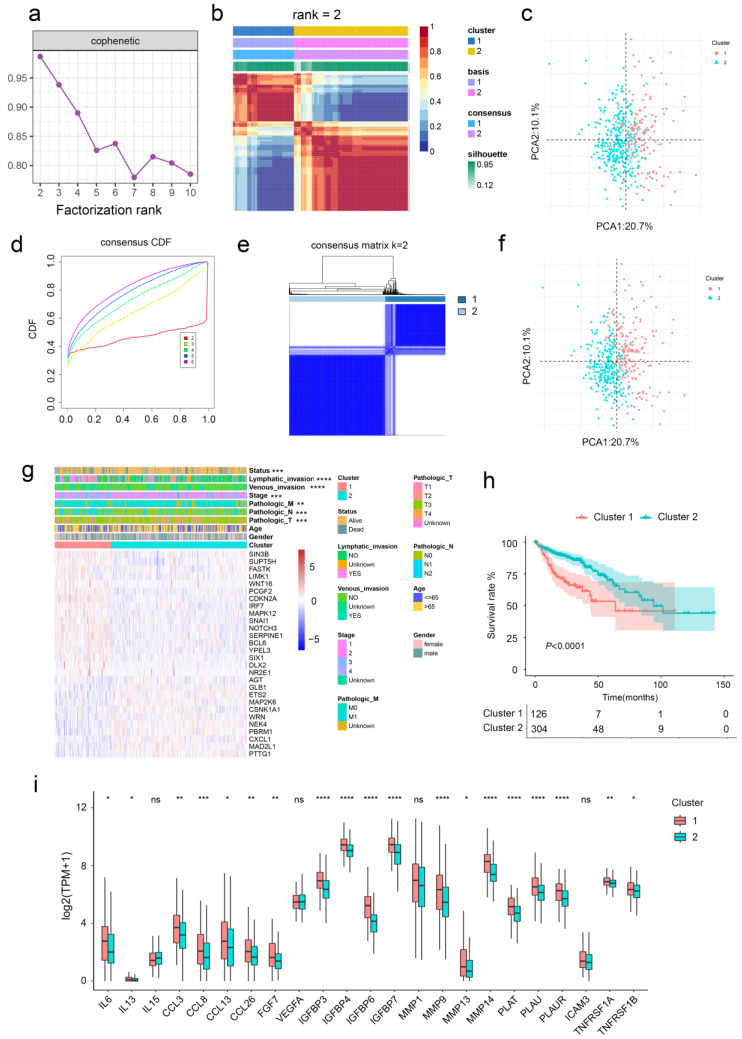
Unsupervised clustering of aging-related genes in CRC. (**a**,**b**) CRC samples were clustered by the NMF method. (**c**) Visualization of NMF results using PCA. (**d**,**e**) CRC samples were clustered by consensus clustering method. (**f**) Visualization of consensus clustering results using PCA. (**g**) The relationships between the clinicopathological features and aging-related gene subgroups. (**h**) Kaplan−Meier curves for OS of TCGA cohort with the aging classes. (**i**) Gene expression of SASP gene sets between two distinct clusters. *, *p* < 0.05; **, *p* < 0.01; ***, *p* < 0.001; ****, *p* < 0.0001; ns, not significant.

**Figure 2 ijms-24-01516-f002:**
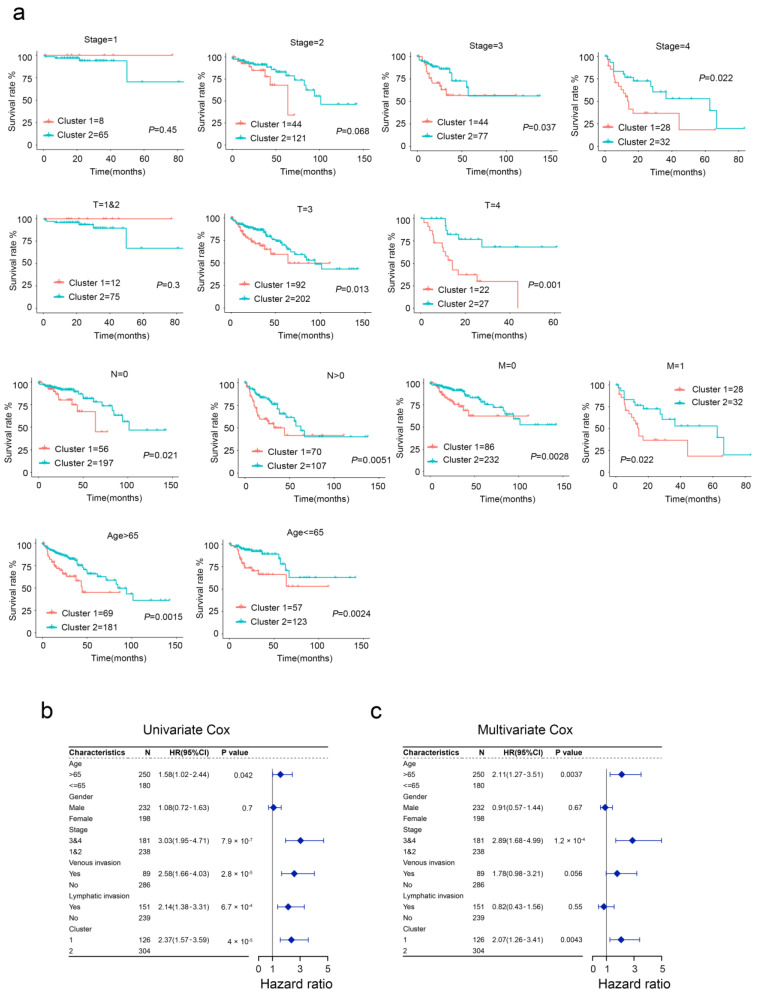
Clinical significance of aging subtypes in the TCGA cohort. (**a**) Survival curve analysis between different subtypes. (**b**,**c**) Univariate and multivariate Cox regression analyses of aging subtypes and other clinical factors.

**Figure 3 ijms-24-01516-f003:**
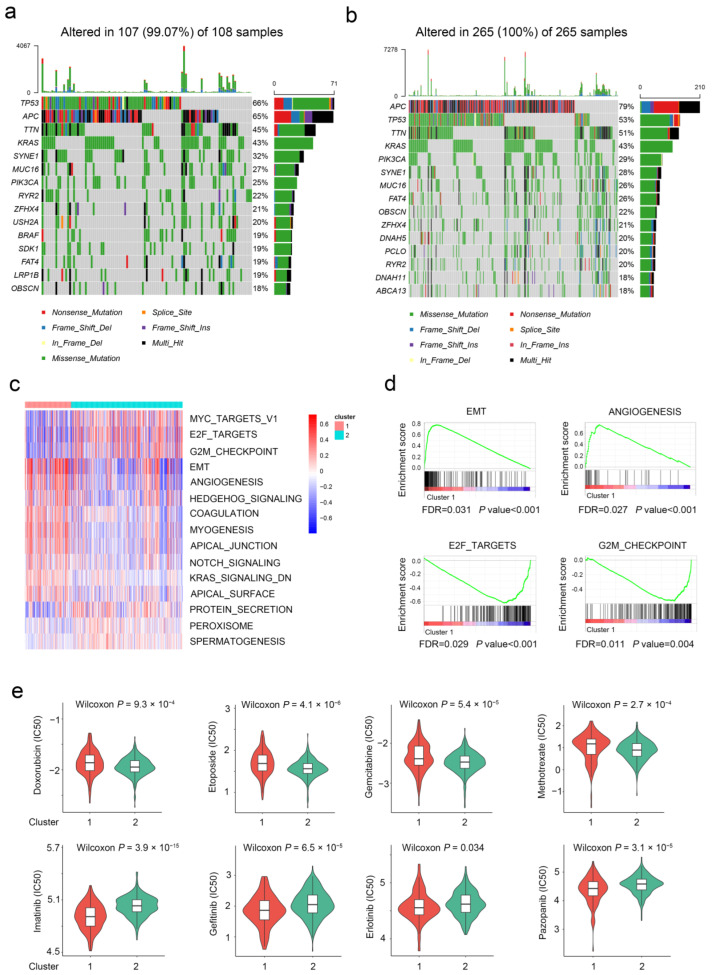
Molecular characteristics of different aging subgroups. (**a**) Landscape of genomic aberrations of cluster 1. (**b**) Landscape of genomic aberrations of cluster 2. (**c**,**d**) GSVA and GSEA analyzed the biological pathways of two aging subtypes. (**e**) Eight common therapeutic drugs with differential IC50 between two aging subtypes.

**Figure 4 ijms-24-01516-f004:**
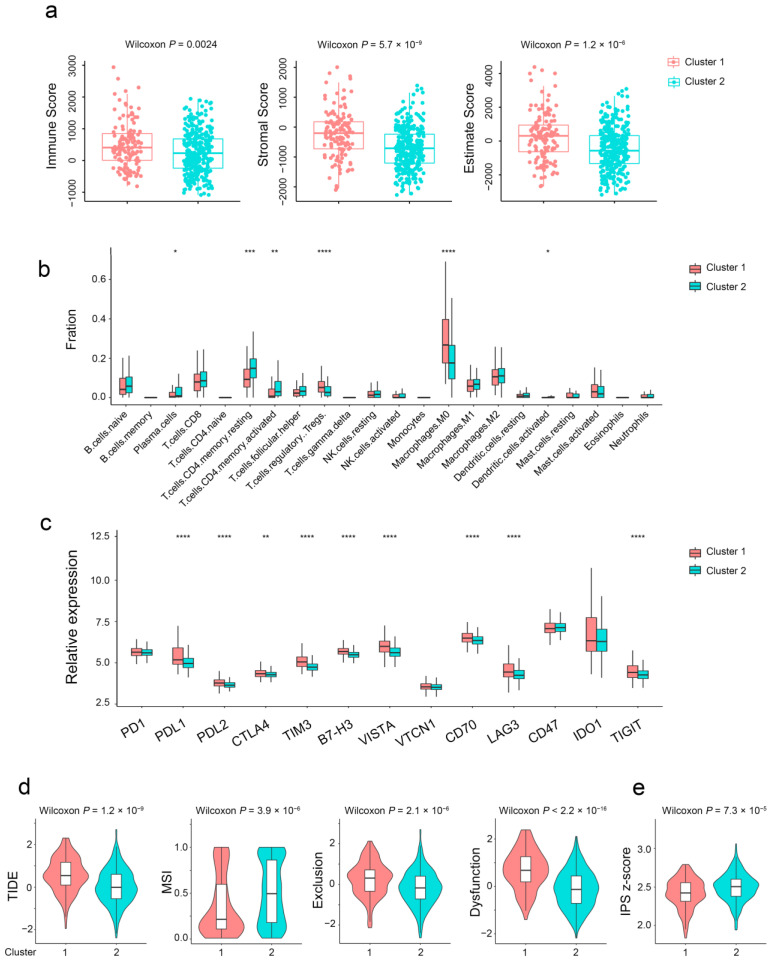
Immune characteristics of different aging subgroups. (**a**) Box plots of the immune, stromal, and estimate score between different aging subtypes. (**b**) Relative proportions of 22 immune cells in different aging subgroups. (**c**) Gene expression of immune checkpoints between two distinct clusters. (**d**,**e**) Violin plot of the TIDE and IPS score between two aging subtypes. *, *p* < 0.05; **, *p* < 0.01; ***, *p* < 0.001; ****, *p* < 0.0001.

**Figure 5 ijms-24-01516-f005:**
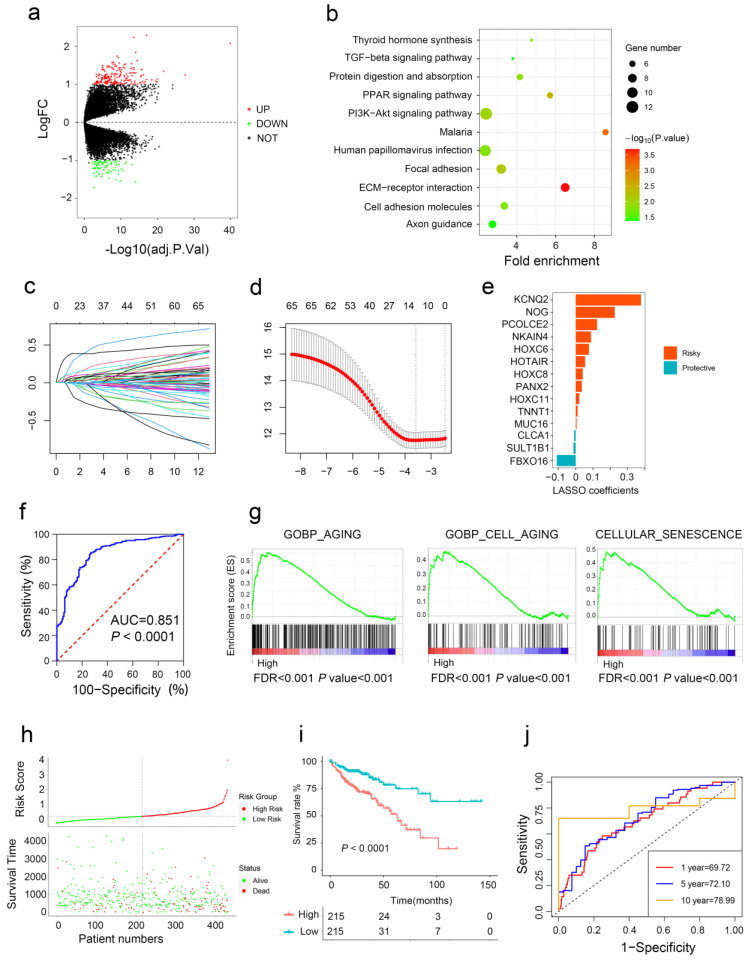
Construction of aging-related risk score to predict CRC prognosis. (**a**) Volcano plots show DEGs between the two subtypes. (**b**) Bubble chart shows the functional enrichment results of DEGs. (**c**,**d**) Aging-related risk score models were constructed using least absolute shrinkage and LASSO methods. (**e**) LASSO coefficients of the 14 aging-related genes. (**f**) ROC curves for predicting aging subtypes by risk score. (**g**) GSEA of aging pathway in high- and low-risk groups. (**h**,**i**) The association of high-risk score with prognosis of CRC patients. (**j**) ROC curves for predicting 1-, 5-, and 10-year OS by risk score.

**Figure 6 ijms-24-01516-f006:**
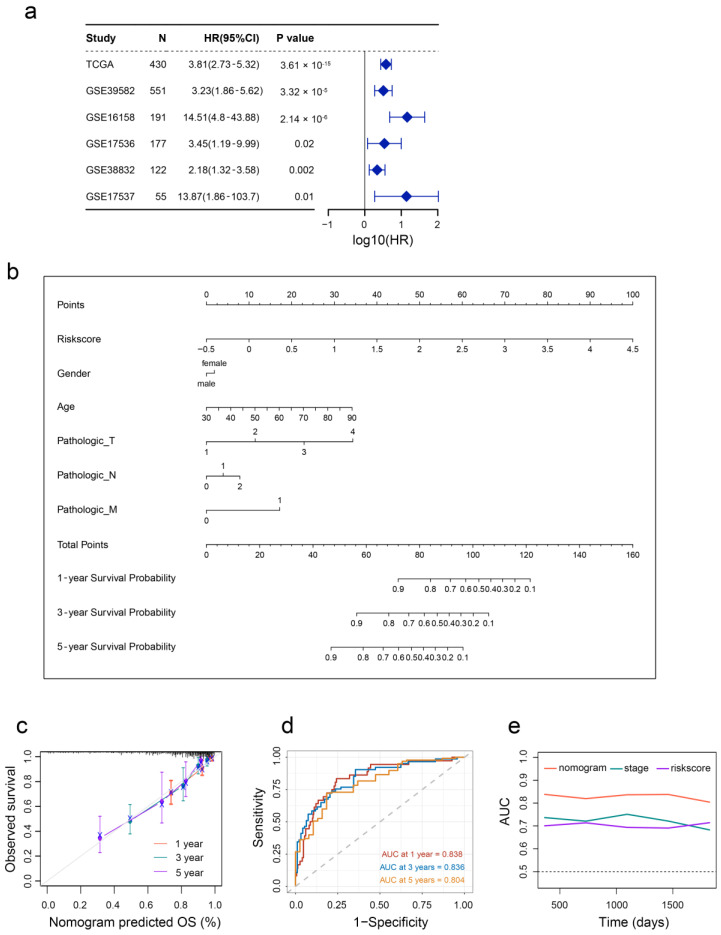
Developed nomogram to predict the probability of survival in CRC patients. (**a**) Univariate Cox regression analyses of risk scores in multiple independent cohorts. (**b**) Prognostic nomogram based on risk scores and clinicopathological characteristics in CRC patients. (**c**) Calibration curves. (**d**) Time-dependent ROC curves at 1, 3 and 5 years. (**e**) Line graph of the area under the curve.

**Figure 7 ijms-24-01516-f007:**
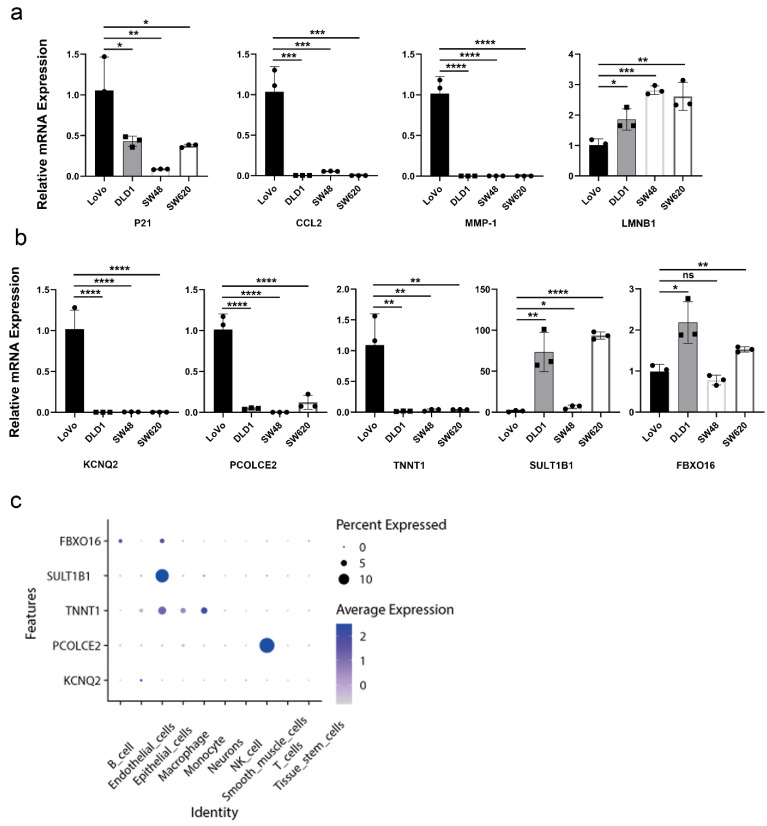
Validation of aging-related genes. (**a**) mRNA expression levels of cellular senescence markers. (**b**) mRNA expression levels of KCNQ2, PCOLCE2, TNNT1, SULT1B1, and FBXO16. (**c**) Dot plot represented the expression of FBXO16, SULT1B1, TNNT1, PCOLCE2, and KCNQ2 of each cell type in CRC tissues. *, *p* < 0.05; **, *p* < 0.01; ***, *p* < 0.001; ****, *p* < 0.0001; ns, not significant.

**Table 1 ijms-24-01516-t001:** Correlation of two cluster groups with clinicopathological parameters based on TCGA CRC data.

Clinical Features	N	Cluster 1	Cluster 2	*p* Value
Age (years)				
≤65	180	57	123	0.391
>65	250	69	181	
Gender				
Female	198	59	139	0.915
Male	232	67	165	
Status				
Alive	336	85	251	0.0008
Dead	94	41	53	
Lymphatic invasion				
Yes	151	68	83	<0.0001
No	239	47	192	
Venous invasion				
Yes	89	40	49	0.0001
No	286	66	220	
Pathologic stage				
Stage1 + 2	238	52	186	<0.0001
Stage3 + 4	181	72	109	
Metastasis				
M0	318	86	232	0.0034
M1	60	28	32	
Lymph node status				
N0	253	56	197	0.0002
N1 + 2 + 3	177	70	107	
Tumor stage				
T1 + T2	87	12	75	0.0007
T3 + T4	320	103	217	

## Data Availability

Data are available in a public, open access repository.

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
