# Peer review of "Identification of Aging and Young Subtypes for Predicting Colorectal Cancer Prognosis and Immunotherapy Responses"

_ijms, 2023, doi:10.3390/ijms24021516_

Round 1

Reviewer 2 Report

Dear Authors,

Thank you for your submission.

Comments:

1. As we know, the consensus molecular subtype (CMS) of colorectal cancer is currently one of the most important prognostic classifications for these neoplasms. Please add a brief description for CMS in your introduction and also explain any genetic mutation differences or overlaps between your suggested aging subgroups and this classification in the discussion section.

2. Please move the methods section before the results.

3.  Graph (g) of figure 1 is vague. To illustrate the relationships between the clinicopathological features and aging-regulated gene subgroups, adding a table with statistical significance is more appropriate.

4. In Kaplan–Meier curves, the time has been defined by days. It is usually defined by months or years. 

Round 2

Reviewer 2 Report

Dear Authors,

Thank you for your revision.